# Intensive Care Unit admission and long-term survival in older patients after elective major noncardiac surgery: A secondary analysis

**Yu-Jia Wu**[1⊘], **Ya-Wei Li**[1⊘], **Na-Ping Chen**[1], **Mo Li**[1], **Ya-Ting Du**[1,2], **Wei-Jie Zhou**[1], **Cong Fu**[1], **Ya-Fei Liu**[1], **Dong-Xin Wang**[1,3*]

1 Department of Anesthesiology, Peking University First Hospital, Beijing, China, 2 Department of Anesthesiology, Beijing Friendship Hospital, Capital Medical University, Beijing, China, 3 Outcomes Research Consortium, Houston, Texas, United States of America

⊘ These authors contributed equally to this work.
* wangdongxin@hotmail.com, dxwang65@bjmu.edu.cn

## Abstract

### Background

Intensive care unit (ICU) admission after surgery is an important part of perioperative care but was found not effective in reducing perioperative mortality. This study was designed to test our hypothesis that ICU admission in older patients after elective major noncardiac surgery was not associated with improved long-term survival.

### Methods

This was a secondary analysis of database from a previous trial and long-term follow-up. Patients were analyzed according to whether they were admitted to ICU after surgery or not. We used propensity score-matching to balance baseline, perioperative, and long-term variables and multivariable Cox proportional hazard regression to adjust confounding factors. Our primary endpoint was overall survival. Secondary endpoints included recurrence-free survival and event-free survival.

### Results

A total of 1712 patients (mean age 69.5 years, 65.3% male, 91.9% cancer surgery) completed long-term follow-up (median 74 months) and were included in this secondary analysis. After propensity score matching, 464 patients remained in the matched cohort, with 232 patients in each group. At the end of follow-up, there were 112 deaths of 232 patients (48.3%) who were admitted to the ICU, compared with 111 deaths of 232 patients (47.8%) who were not: adjusted HR 1.08, 95% CI 0.83–1.41, P = 0.573. Secondary endpoints, including recurrence-free survival and event-free survival, also did not differ significantly between groups.

**Data availability statement:** Data are available from Peking University First Hospital Ethics Committee (contact Dr. Rong-Hui Yu) for researchers who meet the criteria for access to confidential data. Please send the data request to bdyyll@126.com, and we will process your application accordingly.

**Funding:** The study was funded by National Natural Science Foundation of China (No. 82293644; Dong-Xin Wang) and National High Level Hospital Clinical Research Funding (High Quality Clinical Research Project of Peking University First Hospital No. 2022CR78; Dong-Xin Wang). The funders had no role in study design, data acquisition, analysis, interpretation of results, or in the writing and submitting of the report.

**Competing interests:** The authors have declared that no competing interests exist.

**Abbreviations:** ICU, intensive care unit; NYHA classification, New York Heart Association classification; ASA physical status, American Society of Anesthesiologists physical status; MMSE, Mini-Mental State Examination; TNM stage, tumor-node-metastasis stage; RASS, Richmond Agitation Sedation Scale; CAM-ICU,Confusion Assessment Method for the Intensive Care Unit; NYHA, New York Heart Association; NSAIDs, non-steroid anti-inflammatory drugs; HR, hazard ratio; CI, confidence interval.

## Conclusions

For older patients after elective major noncardiac surgery mainly for cancer, ICU admission was not associated with improved long-term survival. Studies are required to identify patients who would benefit from postoperative ICU admission regarding long-term outcomes.

## 1. Introduction

With the global aging population and the development of surgical techniques, an increasing number of older and high-risk patients receive surgical treatment. Intensive care unit (ICU) is an important part of perioperative care for these patients [1]. As an organized system, ICU provides comprehensive vital signs monitoring, intensive treatment, specialized medical care, and multiorgan support, and manage patients with life-threatening illness [2–6]. Indeed, ICU is lifesaving for patients with acute changes in conditions such as those who develop new-onset organ dysfunction after surgery, although at high medical costs [7–10].

However, ICU admission might also produce adverse consequences. For example, patients in the ICU frequently reported negative experiences including pain, anxiety, and sleep disturbances, which might lead to cognitive impairment and delirium [11–13] and even persistent emotional and psychological problems [14,15]. Moreover, ICU survivors had worse long-term survival than the general population [16–18]. A substantial proportion of long-term survivors experienced cognitive decline, functional disabilities, and poor quality of life [15,19–23]. We note that the above studies were mainly conducted in medical ICU patients. In a prospective study involving 44,814 patients from 27 countries, ICU admission after surgery was not associated with improved perioperative mortality [24]. Another population-based analysis reported similar results, i.e., no relationship was found between ICU admission after major surgery and hospital mortality [25]. The rationality of ICU admission after elective surgery thus needs to be further evaluated [26].

A few studies investigated the impact of perioperative ICU admission on long-term outcomes and reported that patients with ICU admission had worse long-term survival; however, some important confounding factors were not adjusted during the analyses [27,28]. In our previous trial, 1802 older patients who underwent major noncardiac surgery were randomized to receive either general anesthesia or combined epidural-general anesthesia [29]; 1712 of them completed long-term follow-up [30]. The purpose of this secondary analysis was to test our hypothesis that ICU admission of older patients after major noncardiac surgery was not associated with improved long-term survival.

## 2. Methods

### 2.1. Study design and ethics

This was a secondary analysis of database from a randomized trial and long-term follow-up [29,30]. The protocol of the underlying trial was approved by the Peking

University Institutional Review Board (IRB) (IRB00001052–11048; principal investigator: D-XW) and registered with the Chinese Clinical Trial Registry (www.chictr.org.cn; ChiCTR-TRC-09000543) and ClinicalTrials.gov (NCT01661907). Written informed consent was obtained from each participant during the underlying trial. The protocol for long-term follow-up was approved by the Clinical Research Ethics Committee of Peking University First Hospital (2014[744]; principal investigator: D-XW) and registered with ClinicalTrials.gov (NCT03012945). Verbal consent was obtained before follow-up data collection.

The protocol of this secondary analysis was approved by the Biomedical Research Ethics Committee of Peking University First Hospital (2024 [639]; principal investigator: D-XW). Because all data came from existing databases and electronic medical record systems and no follow-up contacts with patients or their family members were needed, the Ethics Committee agreed to waive written informed consents. However, all personal data was kept strictly confidential.

## 2.2. Patients

In the underlying trial, we recruited patients aged between 60 and 90 years who were scheduled to undergo elective major noncardiac thoracic and abdominal surgeries with an estimated duration of ≥2 hours. We excluded those who had severe neurological disease, acute myocardial infarction or stroke within 3 months, any contraindications to epidural anesthesia, severe cardiac dysfunction, severe liver dysfunction (Child-Pugh C), or renal failure [29]. During long-term follow-up, we also excluded patients who did not receive intervention or were enrolled twice in the underlying trial [30].

## 2.3. Anesthesia and perioperative management

Intraoperative monitoring was per clinical routine. For patients assigned to general anesthesia alone, anesthesia was induced with midazolam, propofol, and sufentanil and maintained with propofol infusion and/or sevoflurane inhalation, with or without nitrous oxide inhalation, supplemented with opioids. Patient-controlled intravenous analgesia was provided after surgery, which was established with morphine (0.5 mg/ml), programmed to deliver 2-ml boluses with a lockout interval of 6–10 minutes and a background infusion at a rate of 1 ml/h, and used for up to 72 hours.

For patients assigned to combined epidural-general anesthesia, an epidural catheter was inserted before anesthesia induction. Epidural block was confirmed with a test dose of 2% lidocaine and maintained with 0.375–0.5% ropivacaine. General anesthesia was induced and maintained like patients in the general anesthesia group. Patient-controlled epidural analgesia was provided after surgery, which was established with 0.12% ropivacaine and 0.5 µg/ml sufentanil, programmed to deliver 2-ml boluses with a lockout interval of 20 minutes and a background infusion at 4 ml/h, and used for up to 72 hours.

At the end of surgery, patients were usually extubated in the operating room, monitored in the post-anesthesia care unit for at least 30 minutes, and transferred back to general wards. ICU admission after surgery was generally pre-planned and indicated for patients of older age, with severe preoperative comorbidity such as chronic organ dysfunction, and after long-duration and complex surgery. Occasionally, unplanned ICU admission was required for patients with unstable intraoperative conditions, such as massive bleeding or unexpected adverse events. Other perioperative managements were provided according to routine practice.

## 2.4. Baseline and perioperative data collection

Baseline data included demographics, preoperative comorbidities, chronic smoking, alcohol use, and previous anesthesia history. Preoperative laboratory test results included hematocrit, albumin, creatinine, blood glucose, and sodium and potassium levels. General condition was evaluated with the New York Heart Association (NYHA) classification, American Society of Anesthesiologists (ASA) physical status, and Charlson comorbidity index [31]. Cognitive function was evaluated with the Mini-Mental State Examination (MMSE; scores range from 0 to 30, higher score better) [32]. Daily activity was evaluated with the Barthel Index (scores range from 0 to 100, higher score better) [33]. We also collected the site and the

tumor-node-metastasis (TNM) stage of cancer; the latter was classified according to the 8th edition of the American Joint Committee on Cancer staging system [34].

Intraoperative data included type and duration of anesthesia, site, type, and duration of surgery, estimated blood loss, and blood transfusion. At the end of surgery, ICU admission including unplanned ICU admission was recorded. After surgery, patients were visited twice daily (from 8 to 10 am and from 6 to 8 pm) for delirium assessment during the first 7 days or until hospital discharge or death. Sedation or agitation was firstly assessed with the Richmond Agitation Sedation Scale (RASS; scores range from −5 [unable to wake up] to +4 [aggressive], with 0 indicates alertness and calm) [35]. Delirium was then assessed with the Confusion Assessment Method for the Intensive Care Unit (CAM-ICU) [36,37]. Uses of opioids and non-steroid anti-inflammatory drugs (NSAIDs) were documented. Opioid consumption was calculated as sufentanil equivalent [29,30].

From the 8th day after surgery, patients were visited weekly until 30 days after surgery. Discharged patients were contacted via telephone. Other major complications were defined as new-onset medical conditions that were deemed harmful and required therapeutic intervention, i.e., grade II or higher in the Clavien-Dindo classification [38]. All-cause 30-day mortality was reported.

## 2.5. Long-term follow-up and outcomes

Patients and/or their families were followed up once a year via telephone. Data collected during each follow-up contact included reexamination results, treatment for primary surgical disease, hospital readmission, and vital status. Cancer recurrence was defined as reappearance of the same cancer in or near the original place it started. Cancer metastasis was defined as reappearance of the same cancer in another place of the body. The diagnosis of cancer recurrence and/ or metastasis was made by surgeons (and/or radiologists) based on reexamination results. New cancers indicated those with confirmed diagnosis but were different from the original ones. Other major events were defined as any other diseases that required rehospitalization and/or another surgery. The earliest date of each confirmed event was documented [29,30].

The primary endpoint of this study was overall survival, defined as the time interval from index surgery to all-cause death. Secondary endpoints included recurrence-free survival and event-free survival. Recurrence-free survival was defined as the time interval from index surgery to cancer recurrence/metastasis or all-cause death, whichever occurred first. Event-free survival was defined as the time interval from index surgery to cancer recurrence or metastasis, new cancer, other major events, or all-cause death, whichever occurred first.

## 2.6. Statistical analysis

Patients who completed long-term follow-ups were divided into two groups according to whether they were admitted to ICU immediately after surgery or not. The balance of baseline and perioperative variables between the two groups was evaluated with the absolute standard deviation which was defined as the absolute value of differences of means, mean ranks, or proportions divided by the combined standard deviation. Variables with an absolute standardized difference of >0.119 (i.e., $1.96 \times \sqrt{(n1 + n2)/(n1 \times n2)}$) were considered imbalanced between the two groups [39].

Baseline and perioperative variables used for propensity score matching were selected according to clinical importance. Baseline variables included age, sex, body mass index, years of education, comorbidities, chronic smoking, alcohol use, previous anesthesia history, preoperative laboratory test results, NYHA classification, ASA physical status, Charlson comorbidity index, Mini-Mental State Examination score, Barthel index, site and TNM stage of cancer (confirmed by pathological examination results), and study centers. Intra- and postoperative variables included type and duration of anesthesia, site, type, and duration of surgery, estimated blood loss and blood transfusion, use of NSAIDs, sufentanil equivalent, and unplanned ICU admission. Follow-up variables included the time to last follow-up and anticancer therapies (radiotherapy, chemotherapy, reoperation and interventional therapy). A logistic regression model was used to calculate the propensity scores. Patients were matched in a 1:1 ratio using the nearest neighbor matching algorithm with a 0.2

caliper of the propensity score, not allowing the same control patient to match several intervention patients. In sensitivity analysis, we also included major complications within 30 days for propensity score matching, in addition to the above variables.

For primary and secondary endpoints, overall, recurrence-free, and event-free survivals between the two groups of both the full and matched cohorts were analyzed using Kaplan-Meier survival analysis, with differences between groups tested with Log-Rank tests. Univariable Cox proportional hazards models were used to calculate hazard ratio (HR) and 95% confidence intervals (CIs). Multivariable Cox models were used to adjust for pre-determined factors, including age, sex, body mass index, years of education, site of cancer, TNM stage of cancer, type of surgery, and duration of surgery. The Schoenfeld residual method was used to test whether the proportional hazard assumptions were met for the Cox models.

A P value of <0.05 was considered statistically significant. SPSS 26.0 software (IBM SPSS Inc., USA) and R language version 4.4.1 were used for statistical analysis.

## 3. Results

### 3.1. Participants

Between November 21, 2011, and May 25, 2015, 1802 patients were enrolled in the underlying trial. Of these, 82 patients were excluded before intervention, including 49 cancelled surgeries, 25 withdrew consents, and 8 randomization errors. Among the remaining 1720 patients included in long-term follow-ups, 8 patients were excluded due to repeated recruitment for different surgeries; only the first surgery was considered for this analysis. Long-term follow-up ended on May 31, 2020. At last, a total of 1712 patients (91.9% cancer surgery) were included in this secondary analysis, of whom 337 patients required ICU admission after surgery whereas the other 1375 patients did not. After propensity score matching, 464 patients remained in the matched cohort, with 232 patients in each group; after propensity score matching for sensitivity analysis, 438 patients remained in the matched cohort, with 219 patients in each group (Fig 1). The median follow-up duration of all patients was 74 months (interquartile range, 69–88).

### 3.2. Baseline and perioperative data

In the original cohort, when compared with patients who were not admitted to ICU, those who were admitted to ICU after surgery had older age, higher proportions with stroke, hypertension, coronary heart disease, arrhythmia, and diabetes, higher grades of NYHA and ASA class, and higher Charlson comorbidity index, but had lower levels of preoperative hematocrit and albumin, lower MMSE score, and lower Barthel index; they suffered more colorectal, hepatobiliary-pancreatic, and urinary bladder cancers (less lung, esophageal-thymic, and prostate cancers), had more advanced tumor-node-metastasis stages, endured longer anesthesia and surgery, underwent more intraabdominal and open (less intrathoracic and mini-invasive) surgeries, lost more blood, received more blood transfusion, were given more NSAIDs and opioids, and required more unplanned ICU admission after surgery. In the matched cohort, all baseline, perioperative, and long-term follow-up variables were well balanced between the two groups (Table 1).

In the original cohort for sensitivity analysis, patients admitted to ICU developed more major complications within 30 days in addition to the above-mentioned differences. In the matched cohort for sensitivity analysis, all baseline, perioperative (including major complications within 30 days), and long-term follow-up variables were well balanced between the two groups (S1 and S2 Tables).

### 3.3. Perioperative outcomes

In the original cohort, when compared with patients who were not admitted to ICU, those who were admitted to ICU after surgery developed more delirium within 7 days (6.2% [21/337] vs. 2.7% [37/1375], P = 0.001) and more major

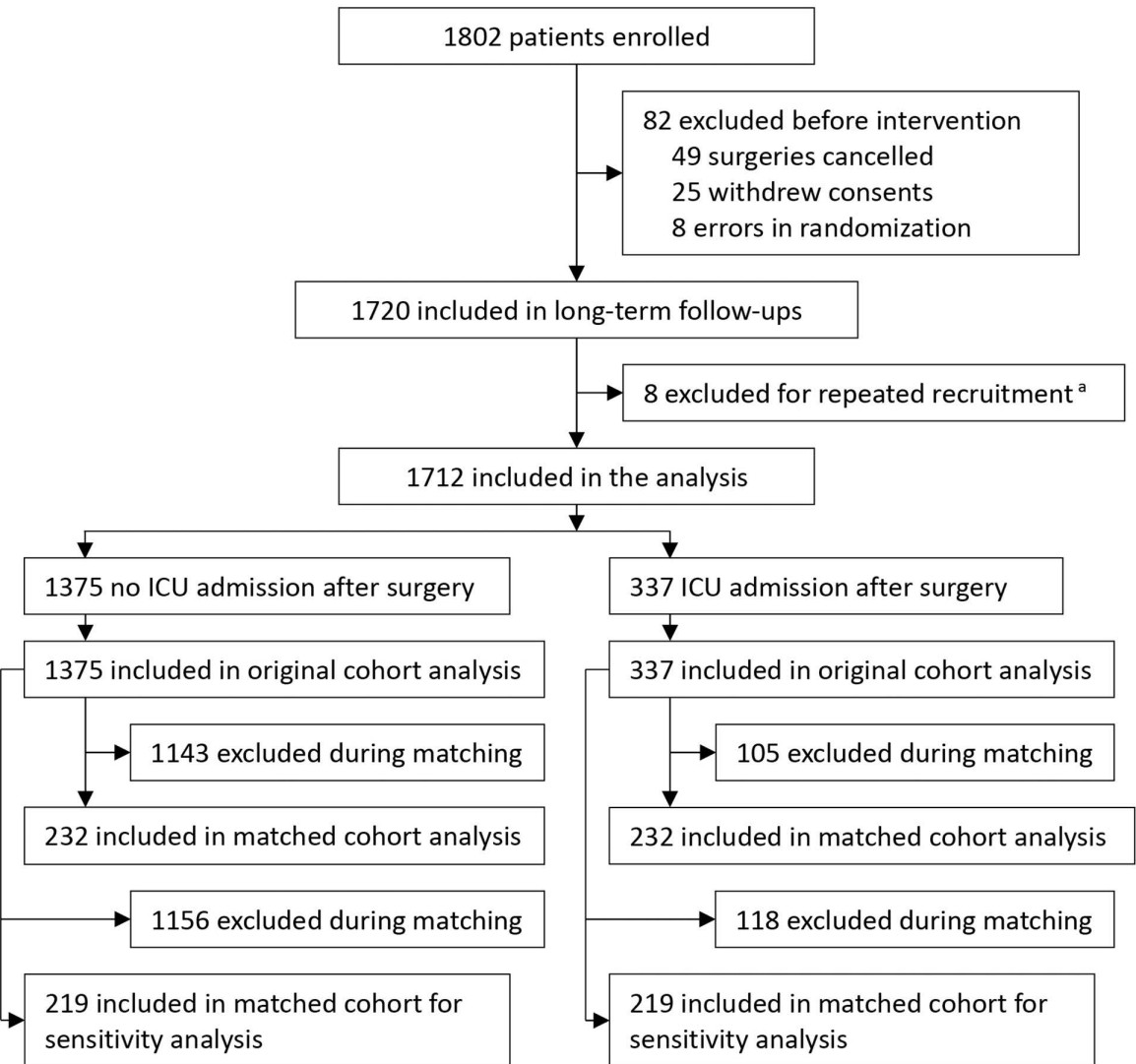

**Fig 1. Flow chart of the study.** ᵃ These patients were enrolled twice for different surgeries. Only their first surgeries were considered for analysis.

complications with 30 days (40.7% [137/337] vs. 18.8% [258/1375], P<0.001), required longer time interval to start oral fluid (median 6 days [interquartile range 4–9] vs. 3 days [1 –6], P<0.001) and food intake (7 days [5 –10 ] vs. 4 days [1 –7], P<0.001), and stayed longer in hospital after surgery (12 days [9 –20 ] vs. 9 days [6 –12 ], P<0.001; Table 2; S3 Table).

In the matched cohort, when compared with patients who were not admitted to ICU, those who were admitted to ICU after surgery developed more major complications with 30 days (35.8% [83/232] vs. 22.0% [51/232], P=0.001), had a longer time before starting oral fluid (6 days [3 –9] vs. 5 days [2 –7], P=0.030) and food intake (7 days [5 –9] vs. 6 days [3 –8], P=0.002), and stayed longer in hospital after surgery (11 days [9 –17 ] vs. 10 days [7 –13 ], P=0.001; Table 2).

In the matched cohort for sensitivity analysis, patients who were admitted to ICU had a longer time before starting oral food intake (7 days [5 –9] vs. 6 days [3 –9], P=0.038). Other perioperative data did not differ between the two groups (S4 Table).

**Table 1. Baseline, perioperative, and long-term follow-up data used for propensity score matching.**

| | All (n = 1712) | Original cohort (n = 1712) | | | Matched cohort (n = 464) | | |
|---|---|---|---|---|---|---|---|
| | | No ICU admission (n = 1375) | ICU admission (n = 337) | ASD | No ICU admission (n = 232) | ICU admission (n = 232) | ASD |
| **Baseline data** | | | | | | | |
| Age, year, mean±SD | 69.5±6.3 | 68.7±5.8 | 73.0±7.4 | **0.586** | 71.7±6.5 | 72.1±7.5 | 0.051 |
| Male sex, n (%) | 1118 (65.3) | 899 (65.4) | 219 (65.0) | 0.008 | 136 (58.6) | 144 (62.1) | 0.072 |
| Body mass index, kg/m², mean±SD | 23.7±3.3 | 23.7±3.2 | 23.5±3.6 | 0.050 | 23.7±3.5 | 23.4±3.6 | 0.066 |
| Education, year, mean±SD | 10±5 | 9±4 | 10±5 | 0.053 | 9±5 | 10±5 | 0.090 |
| Comorbidity, n (%) | | | | | | | |
| Stroke | 85 (5.0) | 51 (3.7) | 34 (10.1) | **0.212** | 19 (8.2) | 17 (7.3) | 0.029 |
| Transient ischemic attack | 23 (1.3) | 15 (1.1) | 8 (2.4) | 0.084 | 4 (1.7) | 4 (1.7) | <0.001 |
| Hypertension | 711 (41.5) | 535 (38.9) | 176 (52.2) | **0.266** | 116 (50.0) | 11 (47.8) | 0.043 |
| Coronary heart disease | 166 (9.7) | 105 (7.6) | 61 (18.1) | **0.271** | 32 (13.8) | 29 (12.5) | 0.034 |
| Arrhythmia | 63 (3.7) | 41 (3.0) | 22 (6.5) | **0.143** | 12 (5.2) | 11 (4.7) | 0.017 |
| Chronic bronchitis | 32 (1.9) | 22 (1.6) | 10 (3.0) | 0.080 | 5 (2.2) | 4 (1.7) | 0.025 |
| COPD | 32 (1.9) | 24 (1.7) | 8 (2.4) | 0.041 | 6 (2.6) | 5 (2.2) | 0.028 |
| Asthma | 27 (1.6) | 19 (1.4) | 8 (2.4) | 0.065 | 6 (2.6) | 4 (1.7) | 0.057 |
| Diabetes | 312 (18.2) | 228 (16.6) | 84 (24.9) | **0.193** | 48 (20.7) | 52 (22.4) | 0.040 |
| Thyroid disease [a] | 45 (2.6) | 38 (2.8) | 7 (2.1) | 0.048 | 7 (3.0) | 6 (2.6) | 0.030 |
| Liver dysfunction [b] | 15 (0.9) | 11 (0.8) | 4 (1.2) | 0.036 | 3 (1.3) | 3 (1.3) | <0.001 |
| Previous cancer [c] | 35 (2.0) | 29 (2.1) | 6 (1.8) | 0.025 | 5 (2.2) | 5 (2.2) | <0.001 |
| Chronic smoking, n (%) [d] | 415 (24.2) | 339 (24.7) | 76 (22.6) | 0.050 | 55 (23.7) | 51 (22.0) | 0.041 |
| Alcohol use, n (%) [e] | 121 (7.1) | 103 (7.5) | 18 (5.3) | 0.095 | 13 (5.6) | 14 (6.0) | 0.019 |
| History of anesthesia, n (%) | 793 (46.3) | 632 (46.0) | 161 (47.8) | 0.036 | 103 (44.4) | 116 (50.0) | 0.112 |
| Laboratory tests | | | | | | | |
| Hematocrit, %, mean±SD | 38.6±5.3 | 38.6±5.1 | 36.2±5.6 | **0.407** | 36.5±5.2 | 36.5±5.6 | 0.008 |
| Albumin, g/L, mean±SD | 40.1±4.4 | 40.5±4.2 | 38.7±4.8 | **0.370** | 39.4±4.6 | 39.1±4.9 | 0.029 |
| Creatinine, μM, mean±SD | 88.1±22.0 | 88.1±21.4 | 88.1±24.2 | 0.001 | 86.1±21.14 | 88.0±23.8 | 0.080 |
| Glucose <4.0 or >10.0 mM, n (%) | 104 (6.1) | 76 (5.5) | 28 (8.3) | 0.101 | 17 (7.3) | 18 (7.8) | 0.016 |
| Na⁺ < 135.0 or >145.0 mM, n (%) | 117 (6.8) | 89 (6.5) | 28 (8.3) | 0.066 | 12 (5.2) | 16 (6.9) | 0.062 |
| K⁺ < 3.5 or >5.5 mM, n (%) | 160 (9.3) | 124 (9.0) | 36 (10.7) | 0.054 | 23 (9.9) | 21 (9.1) | 0.028 |
| NYHA classification, n (%) | | | | **0.279** | | | 0.036 |
| Class I | 1291 (75.4) | 1073 (78.0) | 218 (64.7) | | 156 (67.2) | 160 (69.0) | |
| Class II | 421 (24.6) | 302 (22.0) | 119 (35.3) | | 76 (32.8) | 72 (31.0) | |
| ASA physical status, n (%) | | | | **0.418** | | | 0.068 |
| Class I | 123 (7.2) | 112 (8.1) | 11 (3.3) | | 8 (3.4) | 9 (3.9) | |
| Class II | 1464 (85.5) | 1199 (87.2) | 265 (78.6) | | 191 (82.3) | 195 (84.1) | |
| Class III | 125 (7.3) | 64 (4.7) | 61 (18.8) | | 33 (14.2) | 28 (12.1) | |
| Charlson comorbidity index, point, median (IQR) [f] | 2 (2, 3) | 2 (2, 3) | 2 (2, 3) | **0.280** | 2 (2, 3) | 2 (2, 3) | 0.038 |
| MMSE, point [g] | 29 (27, 30) | 29 (27, 30) | 28 (27, 30) | **0.161** | 29 (27, 30) | 29 (27, 30) | 0.008 |
| Barthel index, point [h] | 100 (100, 100) | 100 (100, 100) | 100 (100, 100) | **0.180** | 100 (100, 100) | 100 (100, 100) | 0.001 |
| Site of cancer, n (%) | | | | **0.698** | | | 0.077 |
| Noncancer | 138 (8.1) | 118 (8.6) | 20 (5.9) | | 16 (6.9) | 14 (6.0) | |
| Gastrointestinal | 179 (10.5) | 139 (10.1) | 40 (11.9) | | 26 (11.2) | 28 (12.1) | |
| Colorectal | 408 (23.8) | 320 (23.3) | 88 (26.1) | | 69 (29.7) | 66 (28.4) | |
| Hepatobiliary-pancreatic [i] | 151 (8.8) | 70 (5.1) | 81 (24.0) | | 37 (15.9) | 41 (17.7) | |

*(Continued)*

| | All (n = 1712) | Original cohort (n = 1712) | | | Matched cohort (n = 464) | | |
|---|---|---|---|---|---|---|---|
| | | No ICU admission (n = 1375) | ICU admission (n = 337) | ASD | No ICU admission (n = 232) | ICU admission (n = 232) | ASD |
| Lung | 271 (15.8) | 253 (18.4) | 18 (5.3) | | 15 (6.5) | 15 (6.5) | |
| Esophageal-thymic | 80 (4.7) | 71 (5.2) | 9 (2.7) | | 8 (3.4) | 8 (3.4) | |
| Reno-ureteral | 148 (8.6) | 127 (9.2) | 21 (6.2) | | 19 (8.2) | 19 (8.2) | |
| Urinary bladder | 173 (10.1) | 128 (9.3) | 45 (13.4) | | 29 (12.5) | 30 (12.9) | |
| Prostate | 136 (7.9) | 132 (9.6) | 4 (1.2) | | 5 (2.2) | 4 (1.7) | |
| Pelvic cavity | 28 (1.6) | 17 (1.2) | 11 (3.3) | | 8 (3.4) | 7 (3.0) | |
| Tumor-node-metastasis stage of cancer [j] | | | | | | | |
| Noncancer, n (%) | 138 (8.1) | 118 (8.6) | 20 (5.9) | 0.112 | 16 (6.9) | 14 (6.0) | 0.036 |
| Tumor stage, n (%) | | | | **0.304** | | | 0.089 |
| $T_x$ | 0 (0.0) | 0 (0.0) | 0 (0.0) | | 0 (0.0) | 0 (0.0) | |
| $T_0$ | 0 (0.0) | 0 (0.0) | 0 (0.0) | | 0 (0.0) | 0 (0.0) | |
| $T_a$ | 12 (0.7) | 9 (0.7) | 3 (0.9) | | 1 (0.5) | 3 (1.4) | |
| $T_{is}$ | 12 (0.7) | 9 (0.7) | 3 (0.9) | | 1 (0.5) | 2 (0.9) | |
| $T_1$ | 314 (18.3) | 268 (19.5) | 46 (13.6) | | 35 (16.2) | 30 (13.8) | |
| $T_2$ | 437 (25.5) | 366 (26.6) | 71 (21.1) | | 47 (21.8) | 52 (23.9) | |
| $T_3$ | 580 (33.9) | 455 (33.1) | 125 (37.3) | | 90 (41.7) | 88 (40.4) | |
| $T_4$ | 219 (12.8) | 150 (10.9) | 69 (20.5) | | 42 (19.4) | 43 (19.7) | |
| Node stage, n (%) | | | | 0.087 | | | 0.068 |
| $N_x$ | 9 (0.5) | 8 (0.6) | 1 (0.3) | | 1 (0.5) | 1 (0.5) | |
| $N_0$ | 1068 (62.4) | 859 (62.5) | 209 (62.0) | | 139 (64.4) | 144 (66.1) | |
| $N_1$ | 262 (15.3) | 206 (15.0) | 56 (16.6) | | 40 (18.5) | 37 (17.0) | |
| $N_2$ | 197 (11.5) | 155 (11.3) | 42 (12.5) | | 32 (14.8) | 30 (13.8) | |
| $N_3$ | 38 (2.2) | 29 (2.1) | 9 (2.7) | | 4 (1.9) | 6 (2.8) | |
| Metastasis stage, n (%) | | | | 0.101 | | | 0.056 |
| $M_x$ | 2 (0.1) | 2 (0.1) | 0 (0.0) | | 0 (0.0) | 0 (0.0) | |
| $M_0$ | 1421 (83.0) | 1141 (82.9) | 280 (83.1) | | 196 (90.7) | 194 (89.0) | |
| $M_1$ | 151 (8.8) | 114 (8.3) | 37 (11.0) | | 20 (9.3) | 24 (11.0) | |
| Study centers, n (%) | | | | 0.034 | | | 0.063 |
| Main center | 1562 (91.2) | 1252 (91.1) | 310 (92.0) | | 208 (89.7) | 212 (91.4) | |
| Sub-centers | 150 (8.8) | 123 (8.9) | 27 (8.0) | | 24 (10.3) | 20 (8.6) | |
| **Intra- and postoperative data** | | | | | | | |
| Type of anesthesia, n (%) | | | | 0.073 | | | 0.095 |
| General | 859 (50.2) | 680 (49.5) | 179 (53.1) | | 124 (53.4) | 113 (48.7) | |
| Combined epidural-general | 853 (49.8) | 695 (50.5) | 158 (46.9) | | 108 (46.6) | 119 (51.3) | |
| Duration of anesthesia, min, median (IQR) | 287 (222, 363) | 278 (217, 350) | 328 (253, 420) | **0.421** | 307 (229, 389) | 307 (240, 397) | 0.017 |
| Site of surgery, n (%) | | | | **0.659** | | | 0.015 |
| Intrathoracic | 402 (23.5) | 373 (27.1) | 29 (8.6) | | 26 (11.2) | 25 (10.8) | |
| Intraabdominal | 1310 (76.5) | 1002 (72.9) | 308 (91.4) | | 206 (88.8) | 207 (89.2) | |
| Type of surgery, n (%) | | | | **0.190** | | | 0.049 |
| Open | 1161 (67.8) | 910 (66.2) | 251 (74.5) | | 158 (68.1) | 163 (70.3) | |
| Thoraco-/laparoscopic | 551 (32.2) | 465 (33.8) | 86 (25.5) | | 74 (31.9) | 69 (29.7) | |
| Duration of surgery, min, median (IQR) | 229 (168, 304) | 222 (160, 290) | 272 (198, 357) | **0.435** | 253 (171, 330) | 251 (186, 338) | 0.025 |

*(Continued)*

**Table 1.** (Continued)

| | All (n=1712) | Original cohort (n=1712) | | | Matched cohort (n=464) | | |
|---|---|---|---|---|---|---|---|
| | | No ICU admission (n=1375) | ICU admission (n=337) | ASD | No ICU admission (n=232) | ICU admission (n=232) | ASD |
| Estimated blood loss, ml, median (IQR) | 100 (50, 300) | 100 (50, 300) | 200 (100, 600) | **0.363** | 200 (50, 400) | 200 (100, 500) | 0.012 |
| Blood transfusion, n (%) | 261 (15.2) | 158 (11.5) | 103 (30.6) | **0.413** | 48 (20.7) | 52 (22.4) | 0.037 |
| Use of NSAIDs, n (%) [k] | 777 (45.4) | 592 (43.1) | 185 (54.9) | **0.238** | 112 (48.3) | 119 (51.3) | 0.061 |
| Sufentanil equivalent, µg [k, l] | 182 (145, 248) | 181 (145, 245) | 191 (150, 272) | **0.179** | 185 (146, 264) | 187 (147, 270) | 0.041 |
| Unplanned ICU admission, n (%) | 10 (0.6) | 3 (0.2) | 7 (2.1) | **0.130** | 2 (0.9) | 3 (1.3) | 0.030 |
| **Long-term follow-up data** | | | | | | | |
| Time to last follow-up, month, median (IQR) | 74 (69, 88) | 74 (68, 88) | 75 (70, 89) | 0.111 | 75 (69, 89) | 75 (70, 81) | 0.065 |
| Anticancer therapy, n (%) | 509 (29.7) | 416 (30.3) | 93 (27.6) | 0.071 | 74 (31.9) | 69 (29.7) | 0.047 |
| Radiotherapy | 75 (4.4) | 62 (4.5) | 13 (3.9) | 0.050 | 11 (4.7) | 13 (5.6) | <0.001 |
| Chemotherapy | 381 (22.3) | 313 (22.8) | 68 (20.2) | 0.075 | 57 (24.6) | 52 (22.4) | 0.052 |
| Reoperation | 73 (4.3) | 59 (4.3) | 14 (4.2) | 0.038 | 7 (3.0) | 11 (4.7) | 0.081 |
| Interventional therapy | 21 (1.2) | 16 (1.2) | 5 (1.5) | 0.044 | 2 (0.9) | 3 (1.3) | 0.038 |

ICU, intensive care unit; COPD, chronic obstructive pulmonary disease; CCI, Charlson Comorbidity Index; NYHA, New York Heart Association; ASA, American Society of Anesthesiologists; ASD, absolute standardized difference.

An ASD of >0.119 is considered imbalanced between the two groups.

[a]Included hyperthyroidism, hypothyroidism, nodular goiter, Hashimoto' s thyroiditis, and thyroid adenoma.

[b]Alanine transaminase and/or aspartate transaminase higher than five times the upper normal limit.

[c]Confirmed by pathologic examination.

[d]Smoking half a pack (10 cigarettes) per day for at least 1 yr, either former or current smoker.

[e]Two drinks or more daily or weekly consumption of the equivalent of 150 ml of alcohol.

[f]According to the Charlson comorbidity index without age.

[g]Scores range from 0 to 30, with higher scores indicating better function.

[h]Scores range from 0 to 100, with higher scores indicating better function.

[i]Included liver, biliary duct, gallbladder, and pancreatic cancer.

[j]According to the American Joint Committee on Cancer 8th Edition Cancer Staging System.

[k]Included those administered intra- and postoperatively (up to 7 days after surgery).

[l]Sufentanil 10 µg (iv) = sufentanil 10 µg (epidural) = fentanyl 100 µg (iv) = remifentanil 100 µg (iv) = morphine 10 mg (iv) = morphine 30 mg (per os) =oxycodone 15 mg (per os) = dezocine 10 mg (iv) = tramadol 100 mg (iv) = pethidine 100 mg (iv).

### 3.4. Long-term outcomes

In the original cohort, patients with ICU admission demonstrated significantly worse overall survival compared to those without. There were 189 deaths of 337 patients (56.1%) who were admitted to ICU after surgery, compared with 496 deaths of 1375 patients (36.1%) who were not: adjusted HR 1.33, 95% CI 1.01 to 1.48, P=0.044 (Fig 2A). In the matched cohort, however, overall survival did not differ between patients with ICU admission and those without. There were 112 deaths of 232 patients (48.3%) who were admitted to the ICU after surgery, compared with 111 deaths of 232 patients (47.8%) who were not: adjusted HR 1.08, 95% CI 0.83 to 1.41, P=0.573 (Fig 2B; Table 3).

In the original cohort, recurrence-free survival was slightly worse in patients with ICU admission, but not statistically significant after adjusting confounding factors. Events occurred in 60.2% (203/337) of patients who were admitted to ICU after surgery, compared with 43.1% (592/1375) of patients who were not: adjusted HR 1.17, 95% CI 0.97 to 1.41, P=0.091 (Fig 2C). In the matched cohort, recurrence-free survival did not differ between patients with ICU admission and

**Table 2. Postoperative outcomes.**

| | All (n = 1712) | Original cohort (n = 1712) | | | Matched cohort (n = 464) | | |
|---|---|---|---|---|---|---|---|
| | | No ICU admission (n = 1375) | ICU admission (n = 337) | P value | No ICU admission (n = 232) | ICU admission (n = 232) | P value |
| Organ support | | | | | | | |
| Mechanical ventilation, n (%) | 181 (10.6) | 0 (0.0) | 181 (53.7) | --- | 0 (0.0) | 122 (52.6) | --- |
| Duration of MV, h, median (IQR) | 0 (0, 0) | 0 (0, 0) | 2 (0, 8) | --- | 0 (0, 0) | 1 (0, 7) | --- |
| LOS in ICU, h, median (IQR) | 0 (0, 0) | 0 (0, 0) | 20 (16, 39) | --- | 0 (0, 0) | 20 (16, 32) | --- |
| Delirium within 7 days, n (%) | 58 (3.4) | 37 (2.7) | 21 (6.2) | **0.001** | 8 (3.4) | 9 (3.9) | 0.805 |
| Major complications within 30 days, n (%) [a] | 395 (23.1) | 258 (18.8) | 137 (40.7) | **<0.001** | 51 (22.0) | 83 (35.8) | **0.001** |
| Time to oral fluid intake, day, median (IQR) | 4 (1, 7) | 3 (1, 6) | 6 (4, 9) | **<0.001** | 5 (2, 7) | 6 (3, 8) | **0.030** |
| Time to oral food intake, day, median (IQR) | 5 (1, 7) | 4 (1, 7) | 7 (5, 10) | **<0.001** | 6 (3, 8) | 7 (5, 9) | **0.002** |
| LOS in hospital, day, median (IQR) | 9 (7, 13) | 9 (6, 12) | 12 (9, 20) | **<0.001** | 10 (7, 13) | 11 (9, 17) | **0.001** |
| All-cause 30-day mortality, n (%) | 8 (0.5) | 5 (0.4) | 3 (0.9) | 0.204 | 2 (0.9) | 1 (0.4) | 0.562 |

ICU, intensive care unit; IQR, interquartile range; LOS, length of stay.

P values in bold indicate <0.05.

[a]See S3 Table for details.

those without. Events occurred in 53.9% (125/232) of patients who were admitted to ICU after surgery, compared with 52.6% (122/232) of patients who were not: adjusted HR 1.12, 95% CI 0.87 to 1.45, P = 0.375 (Fig 2D, Table 3).

In the original cohort, event-free survival was worse in patients with ICU admission. Events occurred in 68.5% (231/337) of patients who were admitted to ICU after surgery, compared with 50.8% (699/1375) of patients who were not: adjusted HR 1.20, 95% CI 1.01 to 1.42, P = 0.040 (Fig 2E). In the matched cohort, event-free survival did not differ between patients with ICU admission and those without. Events occurred in 64.3% (147/232) of patients who were admitted to ICU, compared with 60.3% (140/232) of patients who were not: adjusted HR 1.12, 95% CI 0.88 to 1.42, P = 0.354 (Fig 2F, Table 3).

In the matched cohort for sensitivity analyses, there were no significant differences between patients with and without ICU admission regarding overall (adjusted HR 1.04, 95% CI 0.79 to 1.38, P = 0.770), recurrence-free (adjusted HR 1.07, 95% CI 0.82 to 1.39, P = 0.640), and event-free survivals (adjusted HR 1.10, 95% CI 0.86 to 1.40, P = 0.462; S1 Fig; S5 Table).

## 4. Discussion

Results of this propensity score-matched analysis showed that, in older patients after major noncardiac thoracic and abdominal surgery mainly for cancer, ICU admission was not associated with improved long-term overall, recurrence-free, and event-free survivals. Avoiding ICU admission might be reasonable in relatively "healthy" older patients after elective major surgery.

ICU admission is common in older patients after major surgery. Indeed, recent studies reported rates of ICU admission from 17.5% to 21%; most ICU admission were preplanned [40,41]. In the present study, 19.7% of our patients were admitted to ICU after surgery; only 0.6% required unplanned ICU admission. Main reasons related to ICU admission included baseline (older age, more comorbidities, poorer health status, and higher TNM stages of cancer) and surgery-related (longer duration of surgery and more open surgery) factors. The proportion of ICU admission and the reasons leading to ICU admission in our patients were like those in the literature [40,41].

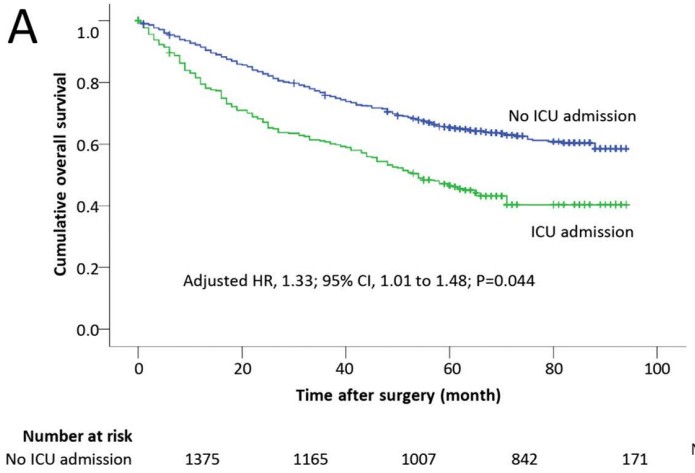

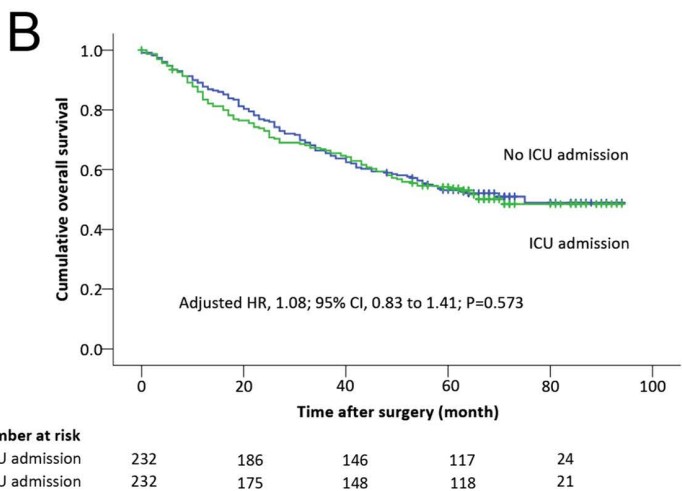

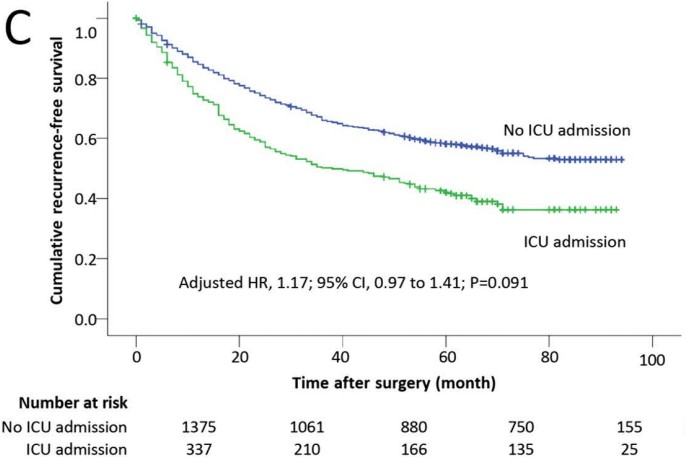

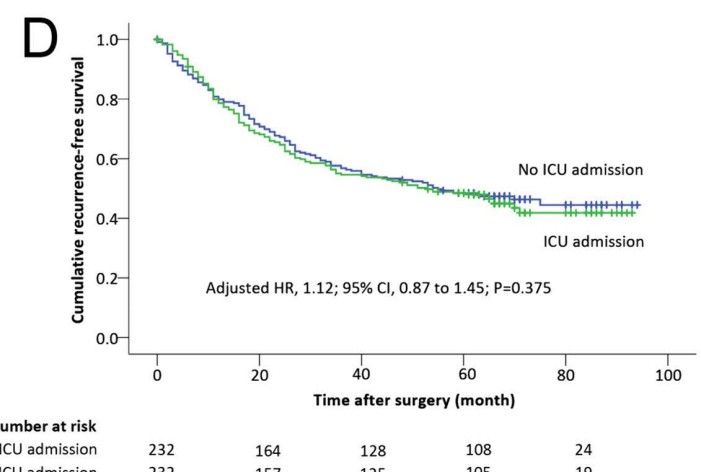

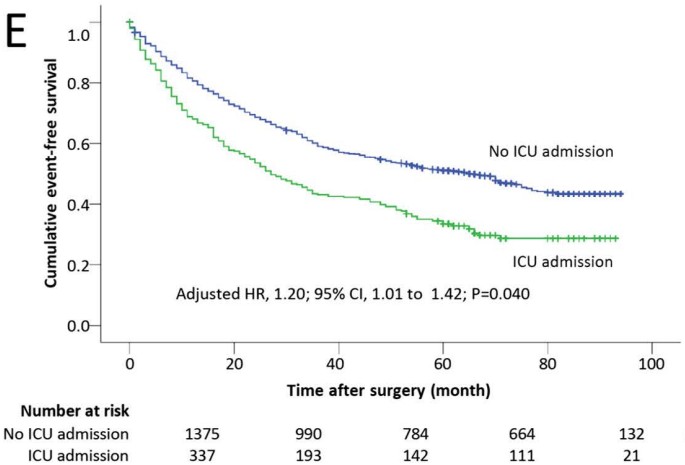

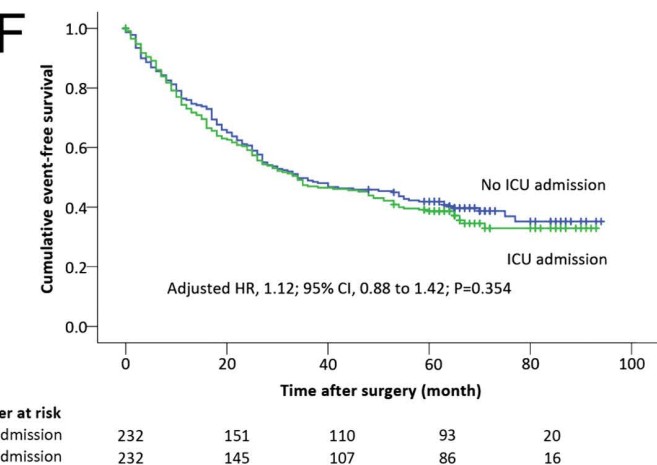

**Fig 2. Kaplan Meier curves for overall (A in original cohort; B in matched cohort), recurrence-free (C in original cohort; D in matched cohort), and event-free (E in original cohort; F in matched cohort) survivals.**

**Table 3. Long-term survival.**

| | Events, n (%) | Unadjusted | | Adjusted | |
|---|---|---|---|---|---|
| | | Hazard ratio (95% CI) [a] | P value | Hazard ratio (95% CI) [b] | P value |
| **Primary endpoint** | | | | | |
| *Overall survival* | | | | | |
| Original cohort (n = 1712) | | | | | |
| No ICU admission (n = 1375) | 496 (36.1) | Ref. | | Ref. | |
| ICU admission (n = 337) | 189 (56.1) | 1.87 (1.58, 2.21) | **<0.001** | 1.33 (1.01, 1.48) | **0.044** |
| Matched cohort (n = 464) | | | | | |
| No ICU admission (n = 232) | 111 (47.8) | Ref. | | Ref. | |
| ICU admission (n = 232) | 112 (48.3) | 1.04 (0.80, 1.35) | 0.782 | 1.08 (0.83, 1.41) | 0.573 |
| **Secondary endpoints** | | | | | |
| *Recurrence-free survival [c]* | | | | | |
| Original cohort (n = 1712) | | | | | |
| No ICU admission (n = 1375) | 592 (43.1) | Ref. | | Ref. | |
| ICU admission (n = 337) | 203 (60.2) | 1.66 (1.42, 1.95) | **<0.001** | 1.17 (0.97, 1.41) | 0.091 |
| Matched cohort (n = 464) | | | | | |
| No ICU admission (n = 232) | 122 (52.6) | Ref. | | Ref. | |
| ICU admission (n = 232) | 125 (53.9) | 1.05 (0.82, 1.35) | 0.699 | 1.12 (0.87, 1.45) | 0.375 |
| *Event-free survival [d]* | | | | | |
| Original cohort (n = 1712) | | | | | |
| No ICU admission (n = 1375) | 699 (50.8) | Ref. | | Ref. | |
| ICU admission (n = 337) | 231 (68.5) | 1.64 (1.42, 1.91) | **<0.001** | 1.20 (1.01, 1.42) | **0.040** |
| Matched cohort (n = 464) | | | | | |
| No ICU admission (n = 232) | 140 (60.3) | Ref. | | Ref. | |
| ICU admission (n = 232) | 147 (64.3) | 1.09 (0.86, 1.37) | 0.489 | 1.12 (0.88 1.42) | 0.354 |

P values in bold indicate <0.05.

[a]Kaplan-Meier survival analysis and log-rank test. Univariable Cox proportional hazards models were used to calculate hazard ratio (HR) and 95% confidence intervals (CIs).

[b]Multivariable Cox proportional hazards model adjusted for age, sex, body mass index, years of education, site of cancer, tumor-node-metastasis stage of cancer, type of surgery, and duration of surgery.

[c]Time interval from index surgery to recurrence, metastasis, or all-cause death, whichever came first.

[d]Time interval from index surgery to recurrence, metastasis, new cancer (confirmed by pathologic examination), other major events (required hospital readmission and/or surgery), or all-cause death, which ever came first.

The rationale of routine ICU admission of older and even high-risk patients after elective major surgery has been questioned, because studies found that ICU admission was not associated with a reduction of perioperative mortality [24–26,41]. We also note that, among our patients, even those who were admitted to ICU after surgery were relatively "healthy", as manifested by the facts that most of them had NYHA class I and ASA class II, and that their median baseline scores of MMSE and Barthel index were 28 and 100, respectively. This is because severely ill patients were excluded during our underlying trial, including those with severe neurological diseases, a history of acute myocardial infarction or stroke within 3 months, severe cardiac dysfunction, severe liver dysfunction (Child-Pugh grade C), or renal failure [29,30].

As can be expected, all-cause 30-day mortality was low and comparable in our patients with and without ICU admission after surgery.

Long-term survival remains poor in patients with ICU stay. In an early observational study, 51% of surgical patients who were admitted to ICU during hospital stay died within 11 years, much higher than a mortality rate of 27% in the age-matched general population [42]. Similar results were reported by a recent cohort study including both medical and surgical patients: hospital survivors following an ICU stay had a 5-year survival of 73.1% and a 10-year survival of 57.5%, compared with expected survivals of 89.5% and 79.1% at equivalent timepoints in the age-matched general population [17]. In the present study, 56.1% of our patients who were admitted to ICU after surgery died during a median 6.3-year follow-up period. Long-term mortality in our patients was much higher than reported results; this could be attributed to the fact that 94.1% of our patients with ICU admission underwent surgery for cancer, whereas most patients in the above studies were admitted to ICU for noncancer diseases [17,42].

Studies investigating the impact of ICU admission on long-term postoperative outcomes were limited. In a prospective cohort study of 1113 adult patients who underwent abdominal surgery, 13% were admitted to ICU after surgery; long-term survival was worse in patients with ICU admission than in age- and gender-matched control patients [27]. In another prospective study of 3348 patients who were hospitalized for cancer (91.9% solid organ cancer), unplanned ICU admission was required in 9.3% patients and was associated with worse 18-month survival [28]. In the above studies, however, reasons leading to ICU admission and some important factors affecting long-term survival (such as cancer stages) were not adjusted during the analysis. The impact on long-term survivals might have been biased by these confounding factors. Our results in the unmatched original cohort also showed that patients with ICU admission had worse overall and event-free survivals.

To investigate the impact of postoperative ICU admission on long-term outcomes, we used propensity score matching to balance the confounding effects of baseline data including site and stage of cancer, perioperative managements including site, type, and duration of surgery, and follow-up variables including anticancer therapies after surgery. In the matched cohort, patients with ICU admission still developed more major complications, required longer time to start oral fluid and food intake, and stayed longer in hospital after surgery; these indicated that some unknown factors related to disease severity were not balanced. In sensitivity analysis, we also included the occurrence of major postoperative complications for propensity score matching to balance the potential impacts of unknown factors. We included major complications for matching because many complications could be attributed to pre- and intraoperative conditions and were not preventable by ICU admission [43]. In the matched cohort for sensitivity analysis, postoperative outcomes were generally comparable between the two groups.

Our results in the matched cohort showed that long-term overall, recurrent-free, and event-free survivals did not differ between patients with and without ICU admission after surgery. Sensitivity analysis confirmed the above results. Potential mechanisms underlying our findings may include the following. (1) Few postsurgical patients in ICU require organ support, which is an important lifesaving technique in ICU [26]. Although 52.6% of our ICU patients required mechanical ventilation, the median duration was only 1 hour, and that of ICU stay was only 20 hours. The role of ICU was therefore more like a post-anesthesia care unit in the present study. In fact, most of our ICU patients were admitted with endotracheal intubation just because they had planned ICU admission. Similar to our situation, a study from Netherlands also reported a median ICU stay of less than one day in patients who were admitted to ICU after elective surgery [44]. In a cohort of ICU patients from both medical and surgical services, 90% of those who survived to hospital discharge did not require mechanical ventilation [45]. (2) Many complications have their roots in pre- and intraoperative conditions and cannot be prevented by ICU admission [43]. For example, surgical complications were caused by surgery and therefore non-preventable by ICU admission; urinary complications were comparable between groups even in the original cohort of our patients. Cardiovascular and respiratory complications were mainly related to baseline conditions and surgical procedures [46,47] and at least partially preventable by careful management. However, although the prevalences of new-onset cardiovascular and respiratory events were high on the day of surgery, most of these complications occurred thereafter [48,49]. A 20-hour ICU stay can only be partially effective, if any, in preventing these complications.

In addition to the observational nature of our study, there are some other limitations. (1) As a secondary analysis, sample size was not estimated to detect survival difference between patients with and without ICU admission after surgery. Nevertheless, with 223 deaths occurred during a 6.3-year period in the matched cohort, our study was well powered to detect a 25% difference between groups. (2) Although we used propensity score matching and multivariable Cox proportional hazard regression during statistical analysis, there might still be unknown and unmeasured confounding factors, as manifested by differences in major complications within 30 days and other postoperative outcomes between the two groups in the matched cohort. In sensitivity analysis, we included major postoperative complications as covariates in the propensity score model for matching; our conclusions were not changed. (3) Studies have shown that frailty has significant impact on long-term survival in ICU patients [50]. As a post-hoc analysis, data of frailty was not collected during our underlying study. Nevertheless, we included all preoperative comorbidities as well as the scores of MMSE and Barthel index as baseline data for propensity score matching; these parameters are important components of modified frailty index [51].

## 5. Conclusions

For older patients after elective major noncardiac surgery mainly for cancer, ICU admission was not associated with improved long-term survival after adjusting confounding factors. Routine ICU admission after surgery could be reasonably avoided in this patient population. More studies are required to identify patients who would benefit from postoperative ICU admission regarding long-term outcomes.

## Supporting information

**S1 Table. Baseline, perioperative, and long-term follow-up data used for propensity score matching (sensitivity analysis).**
(DOCX)

**S2 Table. Individual major complications within 30 days after surgery (sensitivity analysis).**
(DOCX)

**S3 Table. Individual major complications within 30 days after surgery.**
(DOCX)

**S4 Table. Postoperative outcomes (sensitivity analysis).**
(DOCX)

**S5 Table. Long-term survival (sensitivity analysis).**
(DOCX)

**S1 Fig. Kaplan Meier curves for overall (A), recurrence-free (B), and event-free (C) survivals (sensitivity analysis).**
(TIF)

## Acknowledgments

The authors gratefully acknowledge Dr. Jia-Hui Ma (Ph.D., Department of Anesthesiology, Peking University First Hospital) for her help in statistical consultation.

## Author contributions

**Conceptualization:** Dong-Xin Wang.

**Data curation:** Ya-Wei Li, Na-Ping Chen, Mo Li, Ya-Ting Du, Wei-Jie Zhou, Cong Fu, Ya-Fei Liu.

**Formal analysis:** Yu-Jia Wu, Ya-Wei Li, Na-Ping Chen, Mo Li, Dong-Xin Wang.

**Funding acquisition:** Dong-Xin Wang.

**Investigation:** Yu-Jia Wu, Ya-Wei Li, Na-Ping Chen, Mo Li, Ya-Ting Du, Wei-Jie Zhou, Cong Fu, Ya-Fei Liu.

**Methodology:** Yu-Jia Wu, Ya-Wei Li, Dong-Xin Wang.

**Project administration:** Dong-Xin Wang.

**Resources:** Dong-Xin Wang.

**Supervision:** Dong-Xin Wang.

**Validation:** Yu-Jia Wu, Ya-Wei Li, Na-Ping Chen, Mo Li, Ya-Ting Du, Wei-Jie Zhou, Cong Fu, Ya-Fei Liu.

**Writing – original draft:** Yu-Jia Wu.

**Writing – review & editing:** Dong-Xin Wang.

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
