## [Decision Letter · Decision Letter 0]

8 Sep 2025

Dear Dr. Wang,

Thank you for submitting your manuscript to PLOS ONE. After careful consideration, we feel that it has merit but does not fully meet PLOS ONE’s publication criteria as it currently stands. Therefore, we invite you to submit a revised version of the manuscript that addresses the points raised during the review process.

We look forward to receiving your revised manuscript.

Kind regards,

Vincenzo Francesco Tripodi

Academic Editor

PLOS ONE

2. In the online submission form you indicate that your data is not available for proprietary reasons and have provided a contact point for accessing this data. Please note that your current contact point is a co-author on this manuscript. According to our Data Policy, the contact point must not be an author on the manuscript and must be an institutional contact, ideally not an individual. Please revise your data statement to a non-author institutional point of contact, such as a data access or ethics committee, and send this to us via return email. Please also include contact information for the third party organization, and please include the full citation of where the data can be found.

**Comments to the Author**

1. Is the manuscript technically sound, and do the data support the conclusions?

Reviewer #1: Yes

Reviewer #2: Yes

2. Has the statistical analysis been performed appropriately and rigorously?

Reviewer #1: Yes

Reviewer #2: Yes

3. Have the authors made all data underlying the findings in their manuscript fully available?

Reviewer #1: Yes

Reviewer #2: Yes

4. Is the manuscript presented in an intelligible fashion and written in standard English?

Reviewer #1: Yes

Reviewer #2: Yes

Reviewer #1: Dear Authors,

Thank you for submitting your manuscript entitled “Intensive care unit admission and long-term survival in older patients after elective major noncardiac surgery: a secondary analysis” to the PLOS One Journal.

This manuscript reports the findings of secondary analysis of a large randomized trial on elective noncardiac surgery that allocated elderly patients to general vs general with neuraxial anesthesia. The Authors investigated the effect of postoperative Intensive Care Unit (ICU) admission using the propensity score matching statistical technique.

The methodology is sound, and the report of the findings is clear and complete.

My only minor comment would be to state again in the discussion what are the boundaries of the relatively “healthy” older patients that probably don’t benefit of postoperative ICU admission, as you already did in the Methods 2.2 Patients Section and in the original RCT publication (“severe neurologic conditions, acute myocardial infarction or stroke within 3 months, any contraindication for epidural anesthesia, severe heart dysfunction, severe liver dysfunction (Child–Pugh grade C), or renal failure were excluded”).

Repeating this information could benefit the reader by circumscribing the clinical population better.

Kind regards

Reviewer #2: Dear Authors,

thank you for submitting your article entitled “Intensive care unit admission and long-term survival in older patients after elective major non-cardiac surgery: a secondary analysis” to the PLOS One Journal.

This manuscript presents the results of a secondary analysis of a large randomized trial on elective non-cardiac surgery in elderly patients, comparing general anesthesia alone with general anesthesia combined with neuraxial anesthesia. The authors evaluated the impact of postoperative Intensive Care Unit (ICU) admission through propensity score matching.

The methodology is rigorous, and the reporting of results is both clear and comprehensive.

My only minor recommendations are:

-The limitations of the study should be better described in the text;

-I recommend including more discussion to contextualize the reported outcomes within the current literature;

-Is important to reiterate in the Discussion section the specific characteristics of the relatively “healthy” older population that is unlikely to benefit from postoperative ICU admission, as already detailed in Methods Section;

-I suggest refining the English in the text.

Kind regards.

**Do you want your identity to be public for this peer review?** For information about this choice, including consent withdrawal, please see our Privacy Policy

Reviewer #1: **Yes: ** Salvatore Sardo

Reviewer #2: No

---

## [Author Response · Author response to Decision Letter 1]

4 Nov 2025

Reviewer #1:

Thank you for submitting your manuscript entitled “Intensive care unit admission and long-term survival in older patients after elective major noncardiac surgery: a secondary analysis” to the PLOS One Journal.

This manuscript reports the findings of secondary analysis of a large randomized trial on elective noncardiac surgery that allocated elderly patients to general vs general with neuraxial anesthesia. The Authors investigated the effect of postoperative Intensive Care Unit (ICU) admission using the propensity score matching statistical technique.

The methodology is sound, and the report of the findings is clear and complete.

My only minor comment would be to state again in the discussion what are the boundaries of the relatively “healthy” older patients that probably don’t benefit of postoperative ICU admission, as you already did in the Methods 2.2 Patients Section and in the original RCT publication (“severe neurologic conditions, acute myocardial infarction or stroke within 3 months, any contraindication for epidural anesthesia, severe heart dysfunction, severe liver dysfunction (Child–Pugh grade C), or renal failure were excluded”).

Repeating this information could benefit the reader by circumscribing the clinical population better.

Response: Thank you for your valuable suggestion. We clarified the description by repeating the exclusion criteria in the “Discussion” section: “We also note that, among our patients, even those who were admitted to ICU after surgery were relatively “healthy”, as manifested by the facts that most of them had NYHA class I and ASA class II, and that their median baseline scores of MMSE and Barthel index were 28 and 100, respectively. This is because severely ill patients were excluded during our underlying trial, including those with severe neurological diseases, a history of acute myocardial infarction or stroke within 3 months, severe cardiac dysfunction, severe liver dysfunction (Child-Pugh grade C), or renal failure [29, 30].” (page 26, line 17-21; page 27, line 1-3).

Reviewer #2:

Thank you for submitting your article entitled “Intensive care unit admission and long-term survival in older patients after elective major non-cardiac surgery: a secondary analysis” to the PLOS One Journal.

This manuscript presents the results of a secondary analysis of a large randomized trial on elective non-cardiac surgery in elderly patients, comparing general anesthesia alone with general anesthesia combined with neuraxial anesthesia. The authors evaluated the impact of postoperative Intensive Care Unit (ICU) admission through propensity score matching.

The methodology is rigorous, and the reporting of results is both clear and comprehensive.

My only minor recommendations are:

-The limitations of the study should be better described in the text;

Response: Thank you for your comments. We extended the limitation paragraph in the “Discussion” section as below: “…. (2) Although we used propensity score matching and multivariable Cox proportional hazard regression during statistical analysis, there might still be unknown and unmeasured confounding factors, as manifested by differences in major complications within 30 days and other postoperative outcomes between the two groups in the matched cohort. In sensitivity analysis, we included major postoperative complications as covariates in the propensity score model for matching; our conclusions were not changed. (3) Studies have shown that frailty has significant impact on long-term survival in ICU patients [50]. As a post-hoc analysis, data of frailty was not collected during our underlying study. Nevertheless, we included all preoperative comorbidities as well as the scores of MMSE and Barthel index as baseline data for propensity score matching; these parameters are important components of modified frailty index [51].” (page 30, lines 1-11).

-I recommend including more discussion to contextualize the reported outcomes within the current literature;

Response: We have added the following content in the discussion section: “Potential mechanisms underlying our findings may include the following. (1) Few postsurgical patients in ICU require organ support, which is an important lifesaving technique in ICU [26]. Although 52.6% of our ICU patients required mechanical ventilation, the median duration was only 1 hour, and that of ICU stay was only 20 hours. The role of ICU was therefore more like a post-anesthesia care unit in the present study. In fact, most of our ICU patients were admitted with endotracheal intubation just because they had planned ICU admission. Similar to our situation, a study from Netherlands also reported a median ICU stay of less than one day in patients who were admitted to ICU after elective surgery [44]. In a cohort of ICU patients from both medical and surgical services, 90% of those who survived to hospital discharge did not require mechanical ventilation [45]. …” (page 28, lines 21-22; page 29, lines 1-9).

-Is important to reiterate in the Discussion section the specific characteristics of the relatively “healthy” older population that is unlikely to benefit from postoperative ICU admission, as already detailed in Methods Section;

Response: Thank you for your suggestion. We clarified the description by repeating the exclusion criteria in the “Discussion” section: “We also note that, among our patients, even those who were admitted to ICU after surgery were relatively “healthy”, as manifested by the facts that most of them had NYHA class I and ASA class II, and that their median baseline scores of MMSE and Barthel index were 28 and 100, respectively. This is because severely ill patients were excluded during our underlying trial, including those with severe neurological diseases, a history of acute myocardial infarction or stroke within 3 months, severe cardiac dysfunction, severe liver dysfunction (Child-Pugh grade C), or renal failure [29, 30].” (page 26, line 17-21; page 27, line 1-3). Please also see response to Reviewer #1.

-I suggest refining the English in the text.

Response: We have carefully reviewed the entire manuscript and refined the English language. These revisions include correcting grammatical errors and improving sentence structure for clarity and flow, replacing imprecise or informal expressions with standard academic terminology and ensuring consistency in terminology and style throughout the text.

---

## [Editor Report · Decision Letter 1]

20 Nov 2025

Intensive care unit admission and long-term survival in older patients after elective major noncardiac surgery: a secondary analysis

PONE-D-25-19019R1

Dear Dr. Wang,

We’re pleased to inform you that your manuscript has been judged scientifically suitable for publication and will be formally accepted for publication once it meets all outstanding technical requirements.

Kind regards,

Vincenzo Francesco Tripodi

Academic Editor

PLOS ONE
---

## [Editor Report · Acceptance letter]

PONE-D-25-19019R1

PLOS One

Dear Dr. Wang,

I'm pleased to inform you that your manuscript has been deemed suitable for publication in PLOS One. Congratulations! Your manuscript is now being handed over to our production team.

Kind regards,

on behalf of

Dr. Vincenzo Francesco Tripodi

Academic Editor

PLOS One